# BMJ Open Is the growth of the child of a smoking mother influenced by the father's prenatal exposure to tobacco? A hypothesis generating longitudinal study

Marcus Pembrey,[1,2,3] Kate Northstone,[1] Steven Gregory,[1,2] Laura L Miller,[1] Jean Golding[1,2]

▶ Prepublication history and additional material is available. To view please visit the journal (http://dx.doi.org/10.1136/bmjopen-2014-005030).

[1]School of Social and Community Medicine, University of Bristol, Bristol, UK
[2]Centre for Child and Adolescent Health, University of Bristol, Bristol, UK
[3]Institute of Child Health, University College London, London, UK

**Correspondence to**
Professor Jean Golding;
jean.golding@bristol.ac.uk

## ABSTRACT

**Objectives:** Transgenerational effects of different environmental exposures are of major interest, with rodent experiments focusing on epigenetic mechanisms. Previously, we have shown that if the study mother is a non-smoker, there is increased mean birth weight, length and body mass index (BMI) in her sons if she herself had been exposed prenatally to her mother's smoking. The aim of this study was to determine whether the prenatal smoke exposure of either parent influenced the growth of the fetus of a smoking woman, and whether any effects were dependent on the fetal sex.

**Design:** Population-based prebirth cohort study.

**Setting:** Avon Longitudinal Study of Parents and Children.

**Participants:** Participants were residents of a geographic area with expected date of delivery between April 1991 and December 1992. Among pregnancies of mothers who smoked during pregnancy, data were available concerning maternal and paternal prenatal exposures to their own mother smoking for 3502 and 2354, respectively.

**Primary and secondary outcome measures:** Birth weight, length, BMI and head circumference.

**Results:** After controlling for confounders, there were no associations with birth weight, length or BMI. There was a strong adjusted association of birth head circumference among boys whose fathers had been exposed prenatally (mean difference −0.35 cm; 95% CI −0.57 to −0.14; p=0.001). There was no such association with girls (interaction p=0.006). Similar associations were found when primiparae and multiparae were analysed separately. In order to determine whether this was reflected in child development, we examined the relationships with IQ; we found that the boys born to exposed fathers had lower IQ scores on average, and that this was particularly due to the verbal component (mean difference in verbal IQ −3.65 points; 95% CI −6.60 to −0.70).

**Conclusions:** Head size differences concerning paternal fetal exposure to smoking were unexpected and, as such, should be regarded as hypothesis generating.

**Strengths and limitations of this study**

- This study is the first to examine the sex-specific fetal effects of parental prenatal exposure to cigarette smoking when the mother herself smoked during pregnancy.
- Data were collected on a population sample that completed questionnaires blind to the study hypotheses.
- Birth measurements were undertaken using trained staff with repeated validation.
- A variety of sensitivity analyses were undertaken, including separate analyses of primiparae and multiparae, as well as of follow-up of the offspring to determine whether the decrement in birth head circumference was reflected in childhood measurement of IQ. All were in accord with the initial finding.
- The limitation of the study is the failure to obtain comparable data to confirm or negate the study findings.

## INTRODUCTION

Fetal programming via the mother's nutrition and other aspects of her environment is well recognised as a contributor to adult morbidity and mortality[1] and some of these enduring effects are likely to be mediated by epigenetic mechanisms.[2 3] Studies have shown specific DNA methylation patterns in children whose mothers had smoked during pregnancy.[4–7] However, there have been few preconceptional transgenerational studies relating the fetal environment of either parent to the birth outcomes of their own children.

In an earlier study of non-smoking mothers, we found an increase in the birth weight and birth body mass index (BMI) of her sons if she had been exposed in utero to her own mothers' smoking, but there was no

such effect if the study father had been exposed in utero.[8] This lack of paternal influence from his own intrauterine exposure was not unexpected. Indeed, it has been proposed that the paternal line can act as a form of control in studies of maternal effects.[9 10] However, this was not our reason for analysing potential paternal exposure transgenerational effects in our earlier paper[8] or in the present analysis of smoking mothers. They were instigated by studies from Sweden based on samples of individuals born in the town of Överkalix. Their longevity and other health outcomes were linked to detailed historical records of harvests experienced by their ancestors. Although most of the emphasis in the Överkalix study was concerned with exposures in mid-childhood,[11 12] the studies of three cohorts pooled together have demonstrated the effects of exposures of the paternal grandmother (PGM) pre-natally to times of very poor harvests on significantly increased mortality of her granddaughters but not her grandsons.[13] Thus, the presumed effect is from the in utero exposure of the PGM to her son and subsequently to his daughter. Such transgenerational effects are now well supported by rodent experiments showing male-line transmissions and often demonstrating sex-specific transmission on outcomes,[14–17] some focusing on imprinted gene expression in descendants[18] and others on associated epigenetic changes,[19–21] although no transgenerational signal itself has been clearly defined.[22]

Our earlier transgenerational study of intrauterine exposure of non-smoking mothers did not consider relationships with fetal growth if the study mother was also a smoker.[8] Here we use the same cohort, the Avon Longitudinal Study of Parents and Children (ALSPAC), to investigate the fetal growth of offspring of smoking mothers only—comparing the offspring of mothers and fathers who were themselves exposed to cigarette smoke in utero with those who were not exposed in this way. The only study that we are aware of that has looked at an aspect of this question considered the birth weight of the grandchildren comparing those born to mothers who smoked according to whether they themselves had been exposed to their own mothers' smoking in utero.[23] Altogether, they reported a decrease of 70 g if the grandmother had also smoked during pregnancy. The authors did not assess whether this difference was merely a function of variation in the amount smoked by the study mother. Nor did they assess whether there was any effect discernible with the prenatal smoke exposure of the study father, or whether there was any difference between the effects depending on the sex of the offspring.

The current study was therefore carried out to assess whether there is indeed a reduction in the growth of the fetus of a smoking mother if her own mother smoked, and/or whether exposure of the father in utero has any effect on the growth of the child of the smoking mother. In line with the evidence of the accumulating transgenerational human and animal data outlined above, we hypothesise that any effects will differ between boy and girl infants.

## METHODS
### Study sample
The data used in these analyses were collected as part of the ALSPAC, which was designed to assess the ways in which the environment interacts with the genotype to influence health and development.[24] Pregnant women, resident in the study area in south west England with an expected date of delivery between 1 April 1991 and 31 December 1992, were invited to take part. About 80% of the eligible population did so.[25]

Information collected from the study parents during their study pregnancy included details of the maternal and paternal grandparents. Figure 1 illustrates the two pathways of possible influence of parental prenatal exposure to cigarette smoke on the study child that we investigate in this paper.

### The exposures
The women and their partners were sent a number of questionnaires during pregnancy.[26] These elicited information on their current smoking habits and those of their parents (ie, the study grandparents). If they reported that their mothers had smoked, they were asked whether their mothers had smoked when expecting them—and, if so, were given the responses: yes/no/don't know from which to select. Thus, the parents who replied 'don't know' had a mother who smoked but the parent was unsure whether she had smoked during her pregnancy. We have analysed these data in two ways: (A) assuming that all these women did smoke during pregnancy, and (B) omitting the 'don't knows' from the analyses and only analysing those definitely reported as smoking during the study pregnancy (this we have treated as a sensitivity analysis). All mothers who themselves did not smoke during the study pregnancy were excluded from these analyses. Consequently, we compared two groups of grandchildren: those whose grandmothers had smoked during the pregnancy resulting in their parent and whose mothers had also smoked during the pregnancy that resulted in the study child

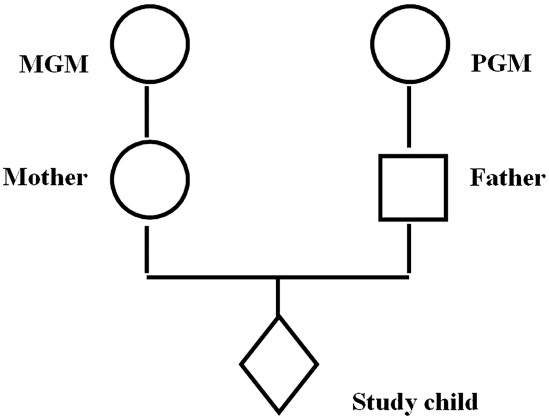

**Figure 1** Diagram of intergenerational linkage, where MGM, maternal grandmother; PGM, paternal grandmother.

**Table 1** The study sample of mothers who smoked during pregnancy

| Number in study | MGM+<br>1781 | MGM−<br>1721 | PGM+<br>1209 | PGM−<br>1145 |
|---|---|---|---|---|
| Maternal smoking in pregnancy (cigarettes per day)* | | | | |
| <10 | 500 (28.0) | 699 (40.6) | 402 (33.3) | 460 (40.2) |
| 10–19 | 811 (45.6) | 698 (40.6) | 524 (43.3) | 475 (41.5) |
| 20+ | 469 (26.4) | 324 (18.8) | 283 (23.4) | 209 (18.3) |
| | p<0.001 | | | p<0.001 |
| Parity* | | | | |
| 0 | 747 (43.3) | 779 (46.1) | 511 (43.3) | 537 (47.8) |
| 1+ | 980 (56.8) | 910 (53.9) | 668 (56.7) | 586 (52.2) |
| | p=0.092 | | | p=0.031 |
| Maternal education level* | | | | |
| CSE or less | 598 (39.6) | 399 (26.3) | 362 (33.8) | 284 (27.6) |
| Vocational | 195 (12.9) | 184 (12.2) | 144 (13.5) | 126 (12.3) |
| O level | 493 (32.6) | 548 (36.2) | 367 (34.3) | 357 (34.7) |
| A level | 181 (12.0) | 291 (19.2) | 156 (14.6) | 195 (19.0) |
| Degree | 45 (3.0) | 93 (6.1) | 41 (3.8) | 66 (6.4) |
| | p<0.001 | | p<0.001 | |
| Gestation (weeks)* | | | | |
| 39+ | 1328 (75.1) | 1279 (74.6) | 860 (71.5) | 868 (75.9) |
| 37/38 | 305 (17.3) | 312 (18.2) | 249 (20.7) | 197 (17.2) |
| <37 | 135 (7.6) | 123 (7.2) | 94 (7.8) | 78 (6.8) |
| | p=0.070 | | p=0.048 | |
| Partner smoking* | | | | |
| No | 478 (29.7) | 487 (30.3) | 341 (28.3) | 344 (30.0) |
| Yes | 1130 (70.3) | 1120 (69.7) | 863 (71.7) | 801 (70.0) |
| | p=0.072 | | p=0.360 | |
| Housing tenure* | | | | |
| Owned/mortgaged | 736 (44.5) | 965 (59.1) | 583 (50.9) | 677 (61.8) |
| Rented public | 628 (38.0) | 406 (24.9) | 386 (33.7) | 247 (22.5) |
| Rented private/other | 291 (17.6) | 263 (16.1) | 177 (15.5) | 172 (15.7) |
| | p<0.001 | | p<0.001 | |
| Maternal alcohol* | | | | |
| Never | 870 (50.9) | 827 (49.6) | 602 (51.7) | 566 (51.5) |
| <1 glass per week | 563 (32.9) | 530 (31.8) | 376 (32.3) | 348 (31.6) |
| 1+ glasses per week | 277 (16.2) | 311 (18.7) | 186 (16.0) | 186 (16.9) |
| | p=0.170 | | p=0.830 | |
| Maternal age (year)† | 25.8 (5.1) | 26.8 (5.1) | 26.1 (5.1) | 26.8 (5.0) |
| | p<0.001 | | p=0.003 | |
| Maternal birthweight (kg)† | 3.12 (0.67) | 3.32 (0.63) | 3.20 (0.65) | 3.22 (0.67) |
| | p<0.001 | | p=0.570 | |

*N (%).
†Mean (SD); + smoked in pregnancy; − did not smoke in pregnancy.
MGM, maternal grandmother; PGM, paternal grandmother.

(maternal grandmother (MGM)+M+ and PGM+M+) with those whose grandmothers had not smoked (MGM−M+ and PGM−M+), respectively. In these analyses, all study mothers smoked during pregnancy. Analyses of fetal growth measures took account of the highest amount smoked by the mother during the study pregnancy, grouped as <10; 10–19; 20+ per day.

### Possible confounders
Other data used in the analyses include the study mother's parity (as ascertained from the maternal report of previous pregnancies resulting in either a live birth or stillbirth, and coded as 0; 1+); gestation (completed weeks 39+; 37–38; ≤36); mother's partner smoking at the start of pregnancy (primarily reported by the partner, but maternal report was used if the partner report was missing: yes; no); maternal age at the birth of the child (continuous); housing tenure as a measure of socioeconomic background (owned or mortgaged; rented public housing; all other), maternal education (highest level of educational attainment—in five levels of increasing achievement) and maternal alcohol consumption when the mother first felt the baby move (not at all; <1 glass per week and one or more glasses per week).

### Outcome measures
At delivery, the baby was weighed to the nearest gram; ALSPAC study staff visited the two main delivery

hospitals each day and measured the crown-heel length and head circumference of available infants in a standardised manner.[24] BMI was calculated as birth weight/length$^2$ (g/m$^2$). In this study, we have used BMI, rather than ponderal index (PI) as our measure of adiposity at birth; although it is traditional to use PI at birth, there is little literature to justify this. It has been suggested that the criteria used to choose whether to use PI or BMI should be a measure that is independent of length.[27] We have assessed which of the two measures is independent of length at each gestation among ALSPAC births and found that BMI satisfies the independence requirement more closely than PI.[8]

Pilot studies before the start of the ALSPAC study had demonstrated that it was usually the student midwife who was given the task of measuring the circumference of the baby's head; she tended to have had little or no training and the measurements made were grossly inaccurate. For this study, we only used measurements that were made by our own staff, after detailed training and with repeated validation over time.

### Statistical analyses

Multivariable linear regression models assessed the study children's adjusted mean birth weight, crown-heel length, head circumference and BMI by parental prenatal smoking exposure. All models were adjusted for parity, maternal education, amount mother smoked, paternal smoking in pregnancy and gestation with MGM−M+ and PGM−M+ as the reference categories. Additional models adjusted for maternal age, housing tenure and maternal alcohol use as well as maternal birth weight.

### Sensitivity analyses

In order to determine consistency of the findings, separate analyses were undertaken for primiparae and multiparae. In order to determine whether the head circumference results were biologically meaningful, we also used the fact that reduced head circumference is associated with lower levels of childhood IQ.[28] Childhood IQ was assessed by trained psychologists at age 8 years (56.2% of eligible children attended), with an abbreviated form of the Wechsler Intelligence Scale for Children (WISC-III).[29] This abbreviated form has been shown to be a valid method for use in research studies.[30]

### RESULTS

In all, there were 3502 births to smoking mothers for whom data were available as to whether their own mothers had smoked when expecting them (table 1). Approximately, half had such a history. Fewer women had information about the prenatal exposures of the father of their study child (n=2354), but again exposure was approximately 50:50.

Comparison of data concerning the potential confounders (table 1) indicates that if either grandmother had smoked prenatally, then the smoking study mother herself was more likely to be a heavy smoker, to have had lower educational attainment and to be younger; in addition, the family was more likely to be living in rented public housing. Not surprisingly, the women who had been exposed in utero (ie, MGM+) had considerably lower mean birth weight themselves (by 199 g) than those not exposed (MGM−). There was no difference in prevalence of smoking by the study father if his own mother had smoked during pregnancy.

Table 2 compares the birth measurements of study children born to parents who had been exposed to smoking in utero. It can be seen that for the women who had themselves been exposed in utero, there was just one statistically significant unadjusted association in their progeny (a lower birth weight for girls), but that this was no longer significant on adjustment. For paternal in utero exposure, however, there were several unadjusted associations (with girls' birth weight and birth length, and with boys' birth length and head circumference). On adjustment, the association with head circumference remained with a 0.35 cm reduction (95% CI −0.57 to −0.14) for boys (p=0.001), but the association for girls was quite different: +0.08 (95% CI −0.11 to +0.28; p for interaction=0.006).

### Sensitivity analyses

The analyses were repeated for primiparae and multiparae separately (see online supplementary tables S1 and S2). The only significant association that remained after adjustment concerned the head circumference of the study sons. The effect sizes were similar for each parity group: for primiparae the effect size was −0.34 cm (95% CI −0.66 to −0.02), p=0.036; for multiparae the adjusted effect size was similar at −0.35 cm (95% CI −0.64 to −0.06), p=0.017. Again there were significant interactions with the sex of the child.

Since this association with head circumference was consistent but unexpected, and since there is evidence that birth head circumference is associated with childhood IQ,[30] we used the same study methodology to assess whether a similar association was apparent between paternal prenatal exposure and childhood IQ. Table 3 demonstrates that there was indeed a reduction in adjusted mean IQ of 2.90 points (95% CI −5.72 to −0.08; p=0.044) for sons of exposed fathers, but no such association for daughters, although the interaction with sex was not statistically significant. Full scale IQ is made up of the sum of two components (performance IQ and verbal IQ) that are, in general, known to have different genetic and environmental components.[31] We therefore have analysed the data to assess whether the associations with PGMs' smoking during pregnancy are associated with one of these components in particular. We found that paternal exposure in utero had a greater effect on his son's verbal IQ (mean adjusted difference −3.65 points; 95% CI −6.60 to −0.70), but with little difference in performance IQ (mean −1.40 points; 95% CI −4.39 to +1.60).

**Table 2** Mean difference $^{(p\ value)}$ (95% CI) in birth measurements of children born to smoking mothers, comparing those where the child's grandmother had smoked with those who had not

| | MGM+ M+ vs MGM− M+ | | PGM+ M+ vs PGM− M+ | |
| | Unadjusted | Adjusted* | Unadjusted | Adjusted* |
|---|---|---|---|---|
| **Birthweight (g)** | | | | |
| Boy | −13 $^{(0.65)}$ (−69 to +43) | −29 $^{(0.24)}$ (−77 to +19) | −55$^{(0.11)}$ (−123 to +13) | −50$^{(0.074)}$ (−104 to +5) |
| Girl | −63$^{(0.022)}$ (−116 to −9) | −31 $^{(0.22)}$ (−81 to +18) | −88$^{(0.010)}$ (−155 to −22) | −11$^{(0.28)}$ (−67 to +45) |
| **Birth length (cm×100)** | | | | |
| Boy | +8 $^{(0.59)}$ (−20 to +36) | −0 $^{(1.00)}$ (−28 to +28) | −37$^{(0.035)}$ (−72 to −3) | −29$^{(0.070)}$ (−61 to +3) |
| Girl | −11$^{(0.44)}$ (−39 to +17) | +7$^{(0.59)}$ (−20 to +35) | −37$^{(0.037)}$ (−71 to −2) | +1$^{(0.96)}$ (−31 to +33) |
| **Head circumference (cm×100)** | | | | |
| Boy | +4 $^{(0.66)}$ (−14 to +23) | −3$^{(0.74)}$ (−22 to +16) | −35$^{(0.003)}$ (−59 to −12) | −35$^{(0.001)}$ (−57 to −14) |
| Girl | −9$^{(0.28)}$ (−27 to +8) | −3$^{(0.76)}$ (−19 to +14) | −17$^{(0.107)}$ (−39 to +4) | +8$^{(0.39)}$ (−11 to +28) |
| **BMI ((kg/m$^2$)×10)** | | | | |
| Boy | −0.3 $^{(0.73)}$ (−1.9 to +1.3) | −0.8 $^{(0.37)}$ (−2.5 to +0.9) | −1.2$^{(0.24)}$ (−3.1 to +0.8) | −1.0$^{(0.31)}$ (−2.9 to +0.9) |
| Girl | −1.8$^{(0.41)}$ (−3.5 to −0.1) | −1.3$^{(0.15)}$ (−3.1 to +0.5) | −2.1$^{(0.056)}$ (−4.2 to +0.1) | −0.6$^{(0.57)}$ (−2.6 to +1.5) |

The data for birth length and head circumference are given in cm×100 so as to aid viewing.
Values in italics indicate a significance of p<0.05.
*Adjusted for maternal parity, maternal education, partner smoked in pregnancy, gestational length at birth of study child and the amount the mother smoked.
M, mother; MGM, maternal grandmother; PGM, paternal grandmother.

## DISCUSSION

We investigated whether the parents' exposure in utero to their own mothers' smoking was associated with differences in fetal growth among women who smoked in pregnancy, and showed an association between paternal in utero exposure and a reduced head circumference in his sons, but not in his daughters. This was an unexpected finding. A series of sensitivity analyses showed the effect to be almost identical in children born to primiparae and to those born to multiparae. We assessed whether there was confirmatory evidence of an impact on brain size by looking at the IQ of the children. We found a significant reduction in total IQ in 8-year-old boys (but not girls) whose PGM smoked during the pregnancy resulting in the study child's father. The IQ effect size was similar in both parity groups and was still present when birth head circumference was taken into account (data not shown). We showed a stronger association with verbal IQ than performance IQ. To the best of our knowledge, there have been no previous studies that have considered any effects of paternal exposure to smoking in utero on his offspring.

### Strengths and limitations

There are a number of limitations to this study: (1) details of smoking of parents and grandparents depend on parental self-report—however, there is considerable information to indicate that adults are unlikely to lie about smoking habits, especially when using anonymised self-completion questionnaires[32]; here we have shown that the mean birth weight of the study mothers who had reported that their own mothers had smoked when they were in utero was 199 g lower than that of those who had reported that their mothers did not smoke at that time, which was about the expected order of difference if the mothers had reported accurately; (2) although the amount the parents smoked was reported, there was no estimate requested for the amount smoked by the grandmothers when pregnant with the study parent—this may have been associated with the outcome, but it is difficult to postulate how such effects might differ between the sexes of the study children; (3) although the ALSPAC study is large, the numbers of women who smoked throughout pregnancy and for whom details are available on the grandmothers' smoking are reduced and consequently the statistical power is relatively low. Among the strengths of this study are the following: (1) it tested a prior hypothesis that early life exposures can have phenotypic effects down the paternal line with sex-specific outcomes; (2) the information on grandparental and parental smoking was collected prior to the birth of the study child, and consequently cannot have been biased by knowledge of fetal size; (3) birth length and head circumference were ascertained by trained measurers using standard techniques, as opposed to the generally inaccurate methods

**Table 3** Mean difference (95% CI) in birth measurements of children born to smoking mothers, comparing those where the child's grandmother had smoked with those who had not

| | MGM+M+ vs MGM−M+ | | | PGM+M+ vs PGM−M+ | |
| --- | --- | --- | --- | --- | --- |
| | Unadjusted | Adjusted* | | Unadjusted | Adjusted* |
| *Total IQ* | | | | | |
| Boys | *−3.87* | −2.45 | | *−4.00* | *−2.90* |
| 95% CI | *(−6.34 to −1.40)* | (−4.96 to +0.07) | | *(−6.92 to −1.08)* | *(−5.72 to −0.08)* |
| p Value | *0.002* | 0.057 | | *0.007* | *0.044* |
| Number | 694 | 612 | | 507 | 482 |
| Girls | *−2.50* | −0.40 | | *−3.03* | −1.36 |
| 95% CI | *(−4.90 to −0.11)* | (−2.85 to +2.05) | | *(−5.78 to −0.28)* | (−4.07 to +1.36) |
| p Value | *0.041* | 0.749 | | *0.031* | 0.327 |
| Number | 617 | 551 | | 456 | 436 |
| *Performance IQ* | | | | | |
| Boys | *−2.64* | −1.48 | | −2.44 | −1.40 |
| 95% CI | *(−5.20 to −0.08)* | (−4.20 to +1.24) | | (−5.40 to +0.50) | (−4.39 to +1.60) |
| p Value | *0.043* | 0.287 | | 0.104 | 0.360 |
| Number | 698 | 616 | | 510 | 485 |
| Girls | −2.46 | −0.50 | | *−3.03* | −1.74 |
| 95% CI | (−5.01 to +0.09) | (−3.19 to +2.19) | | *(−5.88 to −0.18)* | (−4.69 to +1.20) |
| p Value | 0.059 | 0.716 | | *0.037* | 0.245 |
| N | 619 | 552 | | 457 | 436 |
| *Verbal IQ* | | | | | |
| Boys | *−3.75* | −2.40 | | *−4.73* | *−3.65* |
| 95% CI | *(−6.30 to −1.21)* | (−5.00 to +1.20) | | *(−7.81 to −1.66)* | *(−6.60 to −0.70)* |
| p Value | *0.004* | 0.070 | | *0.003* | *0.015* |
| Number | 697 | 615 | | 509 | 484 |
| Girls | −1.98 | −0.15 | | −2.48 | −0.81 |
| 95% CI | (−4.37 to +0.42) | (−2.60 to +2.30) | | (−5.28 to +0.32) | (−3.54 to +1.92) |
| p Value | 0.106 | 0.906 | | 0.082 | 0.561 |
| Number | 617 | 551 | | 456 | 436 |

Values in italics indicate a significance of p<0.05.
*Adjusted for maternal education, parity, partner smoked in pregnancy, gestational length at birth of study child and the amount the mother smoked.
M, mother; MGM, maternal grandmother; PGM, paternal grandmother.

used in most delivery units; (4) IQ was measured using standard methodology by trained psychologists; (5) the study was based on a relatively large population sample, and the results are therefore likely to be generalisable.

## Meaning of the study

Our parallel study of non-smoking mothers looked at the effect of parental exposure in utero; we found the sons were larger at birth (both in regard to birth weight and birth BMI) if the MGM had smoked in the pregnancy that resulted in the study mother. There was no discernible effect of paternal prenatal exposure on the study child's birth weight or BMI; however, there was a slight *increase* in head circumference among the boys born to fathers who had been exposed in utero (mean difference +0.08 cm; 95% CI −0.03 to +0.19).[8]

Attributing the smaller head circumference in boys of smoking mothers to the prenatal exposure of the father through his own mother's smoking raises the question of possible mechanisms. How might the information be transmitted via his sperm or in some other way? As we noted in the introduction to this paper, there is increasing evidence that exposures, especially in early life, can lead to enduring changes in the epigenome that, in turn, can modify gene expression. While transgenerational epigenetic inheritance remains controversial, at least in humans,[33] the phenomenon of genomic imprinting establishes the principle that epigenetic marks such as DNA methylation placed in one generation can influence gene expression in the next. One such imprinted gene is the insulin growth factor 2 (*IGF2*) which is expressed only from the paternally derived chromosome 11, the maternal copy being epigenetically silenced. *IGF2* encodes an endocrine and autocrine/paracrine-acting factor important in directing growth during prenatal development.[34 35] Maternal smoking has been shown to be associated with a 5% higher DNA methylation level at the *IGF2* DMR (differentially methylated region) in the newborn infant,[5] and interestingly in the context of our study, this methylation shift is specific to male offspring. Thus, it is possible that the study father's *IGF2* DMR had been epigenetically modified (including in his fetal testes) by his mother's smoking throughout pregnancy. Furthermore, it is plausible that this epigenetic state could be transmitted via his sperm to the study offspring. Imprinted gene regions

tend to escape the usual widespread erasure of DNA methylation from the paternally derived genome in the preimplantation embryo soon after fertilisation.[36] In support of paternal effects generally, there is a report of hypomethylation at the *IGF2* DMR in umbilical cord blood being associated with paternal obesity suggesting a preconceptional impact of the obesity (and/or exposures related to it) on the reprogramming of imprint marks during spermatogenesis.[37]

## Conclusion

When the mother is a smoker, we found no effect of her own tobacco exposure in utero on the fetal growth of her children. However, when the mother is a smoker, paternal exposure in utero is associated with a reduced head circumference at birth and IQ at 8 years in sons, but not in daughters. We had no prior hypothesis that head circumference would be associated, particularly among sons, so these results must be considered as hypothesis generating, and require testing in further longitudinal data sets.

**Acknowledgements** The authors are extremely grateful to all the families who took part in this study, the midwives for their help in recruiting them, and the whole ALSPAC team, which includes interviewers, computer and laboratory technicians, clerical workers, research scientists, volunteers, managers, receptionists and nurses.

**Contributors** MP and JG had the idea. KN, SG and LLM carried out the statistical analyses. JG and MP wrote the first draft and all authors contributed to the final manuscript.

**Funding** The UK Medical Research Council (MRC), the Wellcome Trust and the University of Bristol currently provide core support for ALSPAC. The statistical analyses for this paper were undertaken with funding from the Medical Research Council (grant no. G1100226).

**Competing interests** None.

**Ethics approval** Ethical approval for the ALSPAC study was obtained from the ALSPAC Law and Ethics Committee and the three Avon-based Local Research Ethics Committees: Bristol and Weston Health Authority: E1808 Children of the Nineties: Avon Longitudinal Study of Pregnancy and Childhood (ALSPAC) (28 November 1989). Southmead Health Authority: 49/89 Children of the Nineties—'ALSPAC' (5 April 1990). Frenchay Health Authority: 90/8 Children of the Nineties. (28 June 1990). Written consent was obtained for all assays of biological samples. Ethics Committees considered voluntarily returned postal questionnaires as implied consent.

**Provenance and peer review** Not commissioned; externally peer reviewed.

**Data sharing statement** ALSPAC is committed to share data with bona fide researchers. See the study website for the conditions of use and access procedures: http://www.bristol.ac.uk/alspac/researchers/.

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
