## [Reviewer comments · BMJ Open]

Some articles will have been accepted based in part or entirely on reviews undertaken for other BMJ Group journals. These will be reproduced where possible.

ARTICLE DETAILS

TITLE (PROVISIONAL)	Is the growth of the child of a smoking mother influenced by the father's prenatal exposure to tobacco? A hypothesis generating longitudinal study
AUTHORS	Pembrey, Marcus; Northstone, Kate; Gregory, Steven; Miller, Laura; Golding, Jean

VERSION 1 - REVIEW

REVIEWER	Zdenka Pausova The Hospital for Sick Children, Toronto, Canada
REVIEW RETURNED	20-Apr-2014

GENERAL COMMENTS	The manuscript reports the results of a large-scale study investigating whether prenatal smoke exposure of either parent influenced the growth of the fetus in a smoking woman, and whether any such effects were dependent on the fetal sex. The study was conducted within the population-based birth cohort – the Avon Longitudinal Study of Parents and Children – involving 3,502 and 2,354 newborns whose mothers smoked during pregnancy and whose data on prenatal exposure of their mother and father, respectively, were available. The fetal growth outcomes studied were body length, body weight, BMI, and head circumference at birth. The analyses were performed while adjusting for potentially confounding effects of maternal parity, maternal education, partner smoking during pregnancy, gestation length and dose of maternal cigarette smoking. The study reports no associations with birth length, weight and BMI, but it finds an association between prenatal exposure of fathers and head circumference of boys but not girls. Similar associations were observed with total IQ and verbal IQ. These are original findings observed in a unique data set. The manuscript is well written. Could the authors comment of the following: 1. In boys whose fathers were exposed prenatally to maternal cigarette smoking, was there a relationship between IQ and head circumference?2. Did the association with head circumference (and IQ) in these boys vary with the dose of maternal cigarette smoking during pregnancy (<10, 10-20, >20)?3. Were similar associations observed in sets of newborns whose (1) both parents were prenatally exposed, (2) mothers only were
--

	prenatally exposed and (3) fathers only were prenatally exposed? 4. In the legends of Tables 2 and 3, do the authors mean “gestation length” by “gestation”?
--	--

REVIEWER	Maritz, Gert University of the Western Cape, Bellville South Africa.
REVIEW RETURNED	29-Apr-2014

GENERAL COMMENTS	The paper makes a valuable contribution in this field of research and open new doors for further research. The information generated in this and follow-up studies can certainly play an important role in educating people to develop a lifestyle that will benefit the health of their progeny in the future. I suggest that: the paragraph on p 17, lines 38 to 50 be rephrased since it can be confusing. the paragraph on p 19, lines 3 to 21 be moved to the methods section of the manuscript. References: Insert a space between the reference number and the actual reference
--

VERSION 1 – AUTHOR RESPONSE

Reviewer 1

The manuscript reports the results of a large-scale study investigating whether prenatal smoke exposure of either parent influenced the growth of the fetus in a smoking woman, and whether any such effects were dependent on the fetal sex. The study was conducted within the population-based birth cohort – the Avon Longitudinal Study of Parents and Children – involving 3,502 and 2,354 newborns whose mothers smoked during pregnancy and whose data on prenatal exposure of their mother and father, respectively, were available. The fetal growth outcomes studied were body length, body weight, BMI, and head circumference at birth. The analyses were performed while adjusting for potentially confounding effects of maternal parity, maternal education, partner smoking during pregnancy, gestation length and dose of maternal cigarette smoking.

The study reports no associations with birth length, weight and BMI, but it finds an association between prenatal exposure of fathers and head circumference in boys but not girls. Similar associations were observed with total IQ and verbal IQ.

These are original findings observed in a unique data set. The manuscript is well written. Could the authors comment on the following:

1. In boys whose fathers were exposed prenatally to maternal cigarette smoking, was there a relationship between IQ and head circumference?

We have now looked at this. For boys born to smoking mothers and whose fathers had been exposed in utero, the correlation between birth head circumference and IQ = 0.217 (n = 115)

2. Did the association with head circumference (and IQ) in these boys vary with the dose of maternal

cigarette smoking during pregnancy (<10, 10-20, >20)?

For those whose mother smoked 1-9 cigarettes/day, $r = 0.10$ ($n = 62$)

For those smoking 10-19 cigarettes $r = 0.39$ ($n = 43$)

There were only 8 children whose mothers smoked 20 cigarettes or more so we have not calculated r for these.

3. Were similar associations observed in sets of newborns whose (1) both parents were prenatally exposed, (2) mothers only were prenatally exposed and (3) fathers only were prenatally exposed? For boys of smoking mothers, the mean head circumference of those whose mothers but not their fathers had been prenatally exposed was identical to those neither of whose parents had been exposed. Both groups for whom the fathers had been exposed in utero had smaller mean head circumferences.

4. In the legends of Tables 2 and 3, do the authors mean "gestation length" by "gestation"? We have changed the footnotes to these two Tables to make the meaning clear.

Reviewer 2

The paper makes a valuable contribution in this field of research and open new doors for further research. The information generated in this and follow-up studies can certainly play an important role in educating people to develop a lofestyle that will benefit the health of their progeny in the future.

I suggest that:

the paragraph on p 17, lines 38 to 50 be rephrased since it can be confusing.

We have rephrased this paragraph to ensure that the meaning is clearer.

the paragraph on p 19, lines 3 to 21 be moved to the methods section of the manuscript.

We have moved that part of the paragraph to the Methods section, and omitted the remainder.

References Insert a space between the reference number and the actual reference

Completed

In response to the suggestions from your editorial assistant:

We had the ethics statement at the beginning of the Methods section of the paper and have now moved this to after the competing interests statement. The funding and data access descriptions are in the acknowledgements section which is part of the main document. The statement on data sharing is that which the ALSPAC organisation requires, but I have amended this slightly. The reader who is interested in access to data is provided with the website and will be able to determine the huge amount of data that is available.